# The Impact of Electroconvulsive Therapy on Negative Symptoms in Schizophrenia and Their Association with Clinical Outcomes

**DOI:** 10.3390/brainsci12050545

**Published:** 2022-04-25

**Authors:** Xiaowei Tan, Donel Martin, Jimmy Lee, Phern Chern Tor

**Affiliations:** 1Department of Mood Disorder and Anxiety, Institute of Mental Health, Singapore 539747, Singapore; tanxiaowei00@gmail.com; 2School of Psychiatry, University of New South Wales, Randwick, NSW 2031, Australia; donel.martin@unsw.edu.au; 3Black Dog Institute, Hospital Road, Randwick, NSW 2031, Australia; 4Department of Psychosis, Institute of Mental Health, Singapore 539747, Singapore; jimmy_lee@imh.com.sg; 5Lee Kong Chian School of Medicine, Nanyang Technological University, Singapore 636921, Singapore; 6Neurostimulation Service, Institute of Mental Health, Singapore 539747, Singapore; 7Duke-NUS Graduate Medical School, National University of Singapore, Singapore 169857, Singapore

**Keywords:** electroconvulsive therapy, schizophrenia, negative symptoms, cognitive function

## Abstract

Objective: The treatment efficacy of electroconvulsive therapy (ECT) for negative symptoms amongst patients with schizophrenia remains unclear. In this study, we aim to examine the effects of ECT on negative symptoms in schizophrenia and their association with other clinical outcomes, including cognition and function. Methods: This is a retrospective data analysis of patients with schizophrenia/schizoaffective disorder treated with ECT at the Institute of Mental Health (IMH), Singapore, between January 2016 and December 2019. Clinical outcomes were assessed by the Brief Psychiatric Rating Scale (BPRS), the Montreal Cognitive Assessment (MoCA), and Global Assessment of Function (GAF). Changes in scores were compared with repeated measures analysis of variance. Sequential structural modelling was utilized to examine the pathway relationships between changes in negative symptoms, global functioning, and cognition functioning after ECT. Results: A total of 340 patients were analysed. Hence, 196 (57.6%), 53 (15.5%), and 91 (26.7%) showed improvements, no change, and deterioration in negative symptoms, respectively. ECT-induced improvement of negative symptoms was significantly associated with improvement of global functioning (direct effect correlation coefficient (*r*): −0.496; *se*: 0.152; *p* = 0.001) and cognition function (indirect effect *r*: −0.077; *se*: 0.037; *p* = 0.035). Moreover, having capacity to consent, more severe baseline negative symptoms, lithium prescription, and an indirect effect of voluntary admission status via consent capacity predicted ECT associated negative symptoms improvement. Conclusion: ECT is generally associated with improvements of negative symptoms in people with schizophrenia, which correlate with improvements of overall function. Possible novel clinical predictors of negative symptom improvement have been identified and will require further research and validation.

## 1. Introduction

Schizophrenia is a heterogenous condition with a variety of natural history trajectories [1,2] that generally shows a pattern of deterioration and increasing treatment resistance with each treatment trial and recurrence of symptoms [3]. Naturalistically, up to 74% of patients with schizophrenia discontinued their antipsychotic treatments within 18 months [4], reflecting the real-world challenge of treating schizophrenia which results in a large economic burden on patients, their families, other caregivers, and society. Electroconvulsive therapy (ECT) is arguably the first among the effective biological methods of treatment for schizophrenia with the potential of augmenting treatment response when used with antipsychotics [5,6]. Even in patients resistant to the gold standard antipsychotic (clozapine), ECT augmentation can result in up to 50% response both in clinical trial [7,8] and real-world settings [9]. In North America [10] and Singapore [11], treatment guidelines recommend the use of ECT for patients with schizophrenia who are not responding well to medication or require rapid response. International reviews [12] and several local studies [13,14] of ECT treatment effects in schizophrenia support the use of ECT in patients with treatment resistant schizophrenia or requiring rapid response, and that ECT augmentation has equivalent to larger symptomatic improvements compared with pharmacotherapy [4] over a much shorter time frame of 2–3 weeks rather than 1–6 months [15]. Current clinical interest in ECT for schizophrenia is primarily for patients with dominant positive symptoms or catatonia [16]. It remains unclear whether ECT is an effective treatment modality for negative symptoms, a major unmet clinical need in schizophrenia [17].

Negative symptoms represent a core component of schizophrenia associated with significant diminution or absence of normal behaviours related to motivation and interest (e.g., avolition, anhedonia, asociality) or expression (e.g., blunted affect, alogia), which accounts for a large part of the poor functional outcome, cognition impairment, and long-term morbidity in patients with this disorder [18,19,20]. Negative symptoms in schizophrenia remain a clinical challenge with small effect sizes and evidence for pharmacological or psychotherapeutic treatment approaches [21,22]. Previous studies have reported that ECT induced a significant improvement of negative symptoms after treatment [23,24,25,26,27,28,29]. However, other studies report no effect or worsening of negative symptoms on average for schizophrenia patients after ECT [30,31,32,33]. Therefore, current treatment strategies of negative symptoms by ECT remain uncertain and it remains largely under explored whether the ECT-induced negative symptom change could contribute to the functional recovery for schizophrenia patients.

In this study, we aimed to examine the effect of ECT on negative symptoms, cognition, and functioning in a retrospective analysis of schizophrenia patients after ECT treatment. Further, we aim to identify predictors of negative symptoms improvement due to ECT treatment.

## 2. Materials and Methods

### 2.1. Study Population

This is a retrospective study of patients with diagnosis of schizophrenia or schizophrenic spectrum disorder and had been prescribed ECT as part of routine clinical care. We included the medical records of patients in the Institute of Mental Health (IMH) in Singapore who were initiated on ECT treatment from January 2016 to December 2019. Ethics approval to conduct the study was obtained from the National Healthcare Group’s Domain Specific Review Board (DSRB No: 2015/01283).

### 2.2. Socio-Demographics and ECT Treatment Modalities

Patients’ sociodemographic and clinical characteristics, including ECT treatment information and outcome assessment, were extracted using the Clinical Alliance and Research in Electroconvulsive Therapy (CARE) Network data collection system as described in our previous studies [34,35,36]. A panel of variables, including age, gender, admission status, consent information, history of episodes and drug resistance, current medicine prescription, and ECT type, have been included in the analysis.

ECT was delivered using a Thymatron system IV device (Somatics, Lake Bluff, IL, USA) or MECTA SpECTrum 5000Q device (MECTA, Tualatin, WA, USA) with handheld electrodes. Each patient’s empirically determined seizure threshold was used for individualized dosing. ECT was given using bitemporal (BT), bifrontal (BF), or right unilateral (RUL) electrode positioning. BT ECT was delivered at 0.5 ms pulse width at 1.5× dose relative to seizure threshold (DRST), BF ECT was delivered at 1.0 ms pulse width at 1.5× DRST, and RUL ECT was delivered at 0.5 ms pulse width at 5× DRST. Propofol (1 mg/kg) and succinylcholine (0.5 mg/kg) were used for anaesthesia and muscle relaxation, respectively.

### 2.3. Clinical Outcome Assessment

The Brief Psychiatric Rating Scale (BPRS) [36] was used to assess psychiatric symptoms before and after ECT treatment. The BPRS is a Likert rating scale ranging from score 1 to 7 in each item. A lower BPRS score indicates a better mental condition. BPRS subscales (positive/psychotic symptoms, negative/withdrawal symptoms, depressive symptoms, and manic/activity symptoms) were calculated according to the factor structure provided by Burlingame G et al. [37]. Specifically, the negative symptoms were measured by a summed score of BPRS item 14. Disorientation, item 16. Blunted affect, item 17. Emotional withdrawal, and item 18. Motor retardation. Intra-class correlation as defined by (MS_rater_ − MS_error_)/(MS_rater_ + (average number of patients per rater − 1) * MS_error_) was 0.78 for BPRS. We used the Montreal Cognitive Assessment (MoCA) [38] to assess cognitive functioning in the language patients were most comfortable with (English, Chinese, Malay, or Tamil). The MoCA is a cognitive screening test designed to assist health professionals in the detection of mild cognitive impairment with lower score indicating worse cognitive function. Patients’ functioning was assessed with the Global Assessment of Functioning (GAF) scale [39]. All assessment scales, including BPRS, MoCA, and GAF, were administered to patients 1–2 days pre-ECT, 1–2 days after 6 sessions ECT or 1–2 days after completion whole course of ECT treatment.

### 2.4. Statistical Analysis

For statistical analysis, all changes in scores were calculated as the post-ECT score minus pre-ECT score. Change in BPRS subscale scores was compared with a paired sample t test. We categorised the change in negative symptoms into three categories: >0 was as “improvement”; <0 as “deterioration”; 0 as “no change”. BPRS total and subscale scores, MoCA and GAF scores at pre-ECT and post-ECT were compared using repeated measures analysis of variance.

For structural equation modelling (SEM) analyses, all categorical variables were dichotomized and dummy coded. Patients with negative symptoms “deterioration” and “no change” were combined into a single category for “no improvement”. Path analysis of ECT clinical outcomes was estimated using maximum likelihood (ML). SEM was used principally for exploratory analysis and hypothesis-generating purposes to explore complex relationships among patients’ characteristics, change in psychiatric symptoms, and other ECT clinical outcomes to identify possible mediation as well as direct and indirect effects of predictors on ECT outcomes [40,41]. Path analysis using SEM is a powerful and commonly used method of statistical analysing that can be used to analyse models that are more complex (and realistic) than multiple regression in psychiatric studies [42,43]. Moreover, path analysis can simultaneously analyse multiple moderators/mediators and compare different models to determine which one best fits the data, which is unlikely to be coped with by multiple regression analysis [44]. Path analysis can also disprove a model that postulates causal relations among variables, although it cannot prove causality. A two-step strategy was adopted in the current path analysis [45,46]. In the first step, we designed the hypothesized pathway models and fitted with our sample. Model fit and misspecification of the initial model was examined using fit indices and modification indices (MI). The decision to explore and keep new pathways was also followed based on their theoretical meaningfulness. The fit of the path models is described using Chi-square (χ^2^) test (*p* value > 0.05 indicates good fit), root mean square error of approximation (RMSE, <0.08 indicates good fit), the comparative fit index (CFI, acceptable fit is indicated by a value ≥ 0.9), the root mean square error of approximation (RMSEA, a value ≤ 0.08 indicates good fit), and standardized root mean square residual (SRMR, ≤0.8 indicates good fit) [41,47]. The second step involved systematically trimming out non-significant pathways and factors, i.e., coefficient estimates with *p*-value > 0.1. At each step, interim evaluations of fit indices and MI were carried out in search of any relevant pathways arising once the model had been simplified.

After working out the initial predictors for ECT clinical outcomes, we constructed a path model and estimated the size and direction of all direct and indirect paths to estimate the mediating role of change in GAF scores in the relationship between change in BPRS subscale symptoms and change in MoCA score. We also constructed a path model to estimate the mediating role of the “consent” variable in the relationship between “admission status” and ECT negative symptom improvement in the outcome prediction analysis as recommended by modification indices (MIs).

The overall process stopped when no additional significant pathways and predictors were suggested by the MI, while all remaining pathways retained statistically significant given acceptable levels of model fit. Both path analysis models were conducted using MPlus, version 7.4.31. Statistical significance for all correlation coefficients was set at *p* < 0.05.

## 3. Results

### 3.1. Participants

A total of 340 patients were included in the present analysis. Patients’ characteristics are listed in Table 1. The average age was 40.9 ± 14.5 years (mean ± SD) and 174 (51.2%) were female. The average number of ECT sessions was 7.2 ± 2.4 (mean ± SD) and the majority of the patients were treated with bifrontal ECT (80.6%). Among the study population, 196 (57.6%) patients presented with an ECT-induced improvement in negative symptoms, 53 (15.5%) without change, and 91 (26.7%) with deterioration in negative symptoms.

### 3.2. Clinical Outcomes

ECT was associated with a significant improvement in all psychiatric symptoms and subscales assessed by BPRS, cognitive function assessed by MoCA (Figure 1, *p* < 0.001 for all measurements), and functioning as assessed by GAF scale. ECT induced a smaller improvement of negative symptoms (1.8 ± 4.2, 20.2% from baseline) compared to psychotic symptoms (6.7 ± 5.6, 41.9%, *p* < 0.001) and manic symptoms (2.3 ± 3.3, 30.7%, *p* = 0.033).

### 3.3. Associations between Symptom Improvement, Cognition and Functioning

Briefly, 270 patients were included into the correlational analyses between changes in symptoms, cognition, and functioning associated with ECT. The results of the path analysis (Table 2 and Figure 2) show that the improvement in psychotic (positive) symptoms (*r* =−0.611, *se* = 0.108, *p* < 0.001), negative symptoms (*r* = −0.496, *se* = 0.152, *p* = 0.001), depressive symptoms *r* = −0.491, *se* = 0.152, *p* < 0.001) and manic symptoms (*r* = −0.681, *se* = 0.196, *p* = 0.001) was directly correlated with improvement in GAF score. Younger age (*r* = −0.084, *se* = 0.04, *p* = 0.034), female (*r* = 2.804, *se* = 1.117, *p* = 0.012), and improvement in GAF score (*r* = 0.155, *se* = 0.057, *p* = 0.007) were directly and significantly correlated with improvement in MoCA total scores. Moreover, GAF significantly mediated the effect of psychiatric symptoms on MoCA (indirect effect: *r* = −0.095, *se* = 0.038, *p* = 0.014 for psychotic symptoms; *r* = −0.077, *se* = 0.037, *p* = 0.035 for negative symptoms; *r* = −0.076, *se* = 0.036, *p* = 0.034 for depressive symptoms; *r* = −0.106, *se* = 0.05, *p* = 0.035 for manic symptoms).

### 3.4. Predictors of Negative Symptoms Improvement

Briefly, 338 patients were included in the prediction model of ECT-induced negative symptom change. The results of the path analysis (Table 3 and Figure 3) show that voluntary admission status (*r* = 0.276, *se* = 0.038, *p* < 0.001) was directly and significantly correlated with capacity to give consent. Capacity to give consent (*r* = 0.135, *se* = 0.06, *p* = 0.025), lithium prescription (*r* = 0.252, *se* = 0.152, *p* = 0.006) and more severe baseline negative symptoms (*r* = 0.067, *se* = 0.006, *p* < 0.001) were directly and significantly correlated with ECT-induced negative symptom improvement. Moreover, capacity to give consent significantly mediated the effect of admission status on ECT negative symptoms response (indirect effect *r* = 0.037, *se* = 0.017, *p* = 0.032).

## 4. Discussion

Negative symptoms represent an unmet clinical need in schizophrenia. Our results support the role of improvement in negative symptoms on neurocognitive and functional recovery in individuals with schizophrenia. Moreover, we identified several factors, including voluntary admission status, own consent, lithium prescription, and more severe baseline negative symptoms, which could predict negative symptom improvement post ECT. These results imply that ECT may be a useful tool for clinicians to treat schizophrenia patients with persistent negative symptoms. To our knowledge, this is the first paper examining the predictors of ECT-induced negative symptoms change.

Similar to previous reports [23,24,25,26,27,28,29], our study found an ECT-induced negative symptom improvement among patients with schizophrenia, but to a smaller extent than the other symptom domains [48,49,50]. Moreover, 26.7% of patients reported worsening of negative symptoms after ECT treatment, an observation that has also been previously reported [30,51]. Similar to previous studies, the interpretation of our results may be biased by patient selection, i.e., the majority of patients treated with ECT in our hospital were due to acute relapse and dominant positive symptoms. Thus, it remains unclear whether ECT is a treatment tool specifically for patients with dominant negative symptoms. Future studies of randomized controlled trials with patients of dominant negative symptoms may help to address this concern.

This study found that the ECT associated negative symptom change, together with all other subdomain symptoms, were significantly associated with patients’ improvement in global functioning. This is consistent with a previous study which demonstrated that various subdomains of psychiatric symptoms were associated with impaired mental/physical functioning, poorer subjective QoL, higher rates of relapse or re-hospitalization, and suicidal ideas etc. among patients with schizophrenia [52]. Among those subdomain symptoms, negative symptoms tend to persist longer than positive symptoms and patients who exhibit significant negative symptoms have particularly poorer functioning in both mental and physical activities [53,54,55]. A prior study with a similar population recruited from our hospital also reported that negative symptoms were significantly correlated with social and occupational functioning in patients with schizophrenia [56].

We used the GAF scale to assess patient functioning. GAF was designed to rate both symptom severity and social/role functioning [57,58]. In our study population, at pre and post ECT, most of these patients are likely to have remained as inpatient. The range of behaviours observed in wards would be limited. This suggests that the ECT-induced change of GAF score was likely contributed by the symptom severity rather than social or role functioning, which might explain our finding that improvements in all the subdomains were significantly associated with GAF change. This may also explain our observation that GAF improvement was associated with improvement of cognitive function, as it reflects the global symptoms change. Indeed, because of this, the GAF scale has often been criticised as a true measurement of functioning in routine clinical settings [57,59,60,61]. Studies have also begun to shift away from GAF to other functional instruments such as The Social and Occupational Functioning Assessment Scale (SOFAS) [62,63,64].

In our study, negative symptoms, together with other symptom subdomains, are significantly correlated with ECT-induced cognitive change, which is consistent with the extant literature in that more severe symptoms (either overall or subdomain psychiatric symptoms) are associated with poor prospective memory, insight, executive functioning, facial perception, facial emotion recognition, emotion processing and perception, social perception, and theory of mind etc. [65,66]. Neurocognitive deficits, a core feature of schizophrenia, have been widely studied and considered the most significant correlate of impaired functioning and QoL in individuals with schizophrenia. Our observation suggests that ECT may be an effective remedy to restore patients’ functioning via an improvement in psychiatric symptoms. Moreover, treatments targeting functional recovery, either for social functioning or cognition functioning among patients with schizophrenia, should focus on all subdomains of psychiatric symptoms, including negative symptoms, in order to ensure optimal results.

We found that the capacity to give consent and more severe baseline negative symptoms could predict negative symptoms improvement. Moreover, patients who were voluntarily admitted were more likely to give own consent for ECT treatment and this indirectly resulted in a better improvement of negative symptoms. It has been widely recognised that individuals with schizophrenia, on average, have significantly greater difficulty than healthy controls with regard to seeking medical treatment and demonstrating adequate capacity for treatment consent due to severe psychiatric symptoms, impaired cognition function, and lack of insight etc. [67,68]. Published literature in real world settings shows that patients with different admission status and decisional capacity may have different ECT treatment outcomes, including objective psychiatric symptoms, QoL, cognition, and functions [69]. In this study, we observed that patients with voluntary admission status have better ECT associated improvement of negative symptoms, which is likely due to better decisional capacity. This is possibly due to the reverse correlation of those patients who had better decision capacity with more severe baseline negative symptoms, which subsequently resulted in a better improvement of ECT-induced negative symptoms. Although the correlation between decision capacity and baseline negative symptoms is statistically non-significant in our study dataset, in many previous reports for subjects with schizophrenia, decisional capacity was significantly and reversely associated with negative symptoms, such as apathy and avolition, but not psychotic symptoms [70,71,72]. Thus, our observation implies that ECT may be an efficient treatment option for schizophrenia patients with more severely impaired negative symptoms.

Among a panel of medicines that had been prescribed for patients with schizophrenia, lithium stood out as a factor that was associated with ECT-induced negative symptom improvement. Lithium is a well-known mood stabilizer that possesses antioxidant and neuroprotective properties [73]. It has been also used as an antiepileptic agent in preclinical and clinical studies. There is limited literature on the effect of lithium in patients undergoing ECT. The mechanism of lithium to be associated with ECT-induced negative symptom outcomes remains unclear but might be due to the neurophysiological effect of lithium via lowering the electroshock induced seizure thresholds [74,75]. The correlation of lithium with negative symptom response may also be attributed to the cognitive disturbance caused by a combination of ECT and lithium prescription [76].

There are several limitations of our study. Negative symptoms were assessed by the BPRS scale, which is a validated brief screening instrument of general psychiatric symptoms, including negative symptoms, but not a measurement tool fully assessing all dimensions of negative symptoms, such as The Scale for the Assessment of Negative Symptoms (SANS). However, as patients referred for ECT treatment were generally very ill and had experienced several rounds of drug treatment, a brief screening tool may be more appropriate than a detailed assessment tool. Moreover, although our SEM model partially replicated the causal pathway initiated by ECT treatment, randomized clinical trials with larger sample sizes are necessary to validate the efficacy of ECT on dominant negative symptoms. Moreover, as our patients are largely comprised of patients with dominant positive psychotic symptoms, the specificity of ECT-induced negative symptom improvement remains unclear. Finally, it is non-negligible that a small proportion of patients experienced ECT-induced negative symptom deterioration. Future studies are needed to clarify the underlying mechanisms and the potential treatment regime for this group of patients with ECT-associated secondary negative symptoms.

## 5. Conclusions

In summary, we found that, in a naturalistic clinical care setting, ECT treatment resulted in general clinical improvements. These symptom improvements contributed to the improvement in global functioning and cognitive function post-ECT. While the effect of ECT on negative symptoms appears smaller on average compared to other symptom domains, we did observe a subgroup of patients with significant negative symptom improvement. The identification and characterization of this group of patients who might benefit from ECT treatment could shed light on the precise utilization of ECT in the treatment of negative symptoms.

## Figures and Tables

**Figure 1 brainsci-12-00545-f001:**
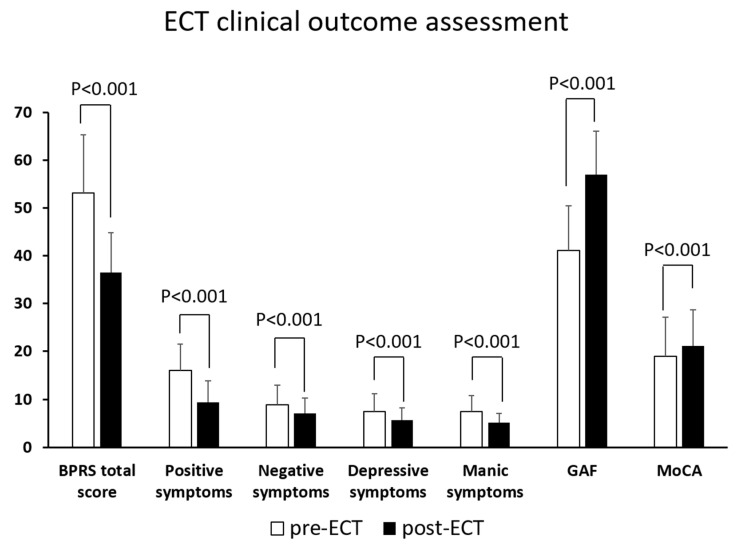
ECT clinical outcome assessment. Abbreviations: SD, standard deviation; BPRS, Brief Psychiatric Rating Scale; GAF, Global Assessment of Functioning; MoCA, Montreal Cognitive Assessment.

**Figure 2 brainsci-12-00545-f002:**
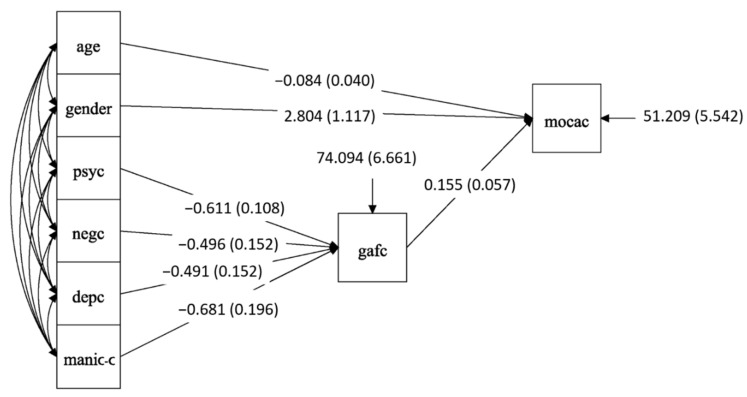
Diagram of pathway correlation among ECT induced negative symptoms change, global functioning and cognition function. Abbreviations: psyc, change of psychotic (positive) symptoms; negc, change of negative symptoms; depc, change of depressive symptoms; manic-c, change of manic symptoms; gafc: change of Global Assessment Functioning; mocac, change of Montreal Cognitive Assessment.

**Figure 3 brainsci-12-00545-f003:**
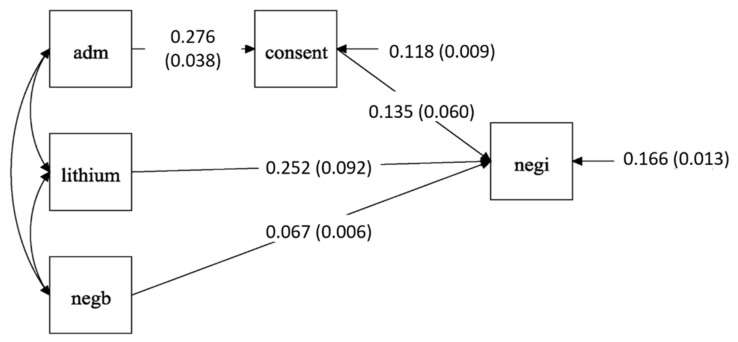
Diagram of pathway correlation among predictors and ECT induced negative symptoms improvement. Abbreviations: adm, admission status; negb, baseline negative symptoms; negi, negative symptoms improvement.

**Table 1 brainsci-12-00545-t001:** Patients’ sociodemographic and clinical characteristics.

Patient Characteristics		Total Sample (*n* = 340)	Negative Symptoms Improvement (196, 57.6%)		Negative Symptoms No Change (53, 15.5%)		Negative Symptoms Deterioration (91, 26.7%)	
Age (years, mean ± SD)		40.9 ± 14.5	39.8 ± 14		42.3 ± 14.5		42.6 ± 15.3	
Number of ECT sessions (mean ± SD)		7.2 ± 2.4	7.3 ± 2.4		6.9 ± 1.9		7.4 ± 2.6	
Gender (N, %)	Female	174	96	55.2%	31	17.8%	47	27.0%
	Male	166	100	60.2%	22	13.3%	44	26.5%
Admission status ^#^ (N, %)	Voluntary	156	90	57.7%	26	16.7%	40	25.6%
	Involuntary	183	105	57.4%	27	14.8%	51	27.9%
Consent ^#^ (N, %)	Own Consent	55	39	70.9%	9	16.4%	7	12.7%
	Consent by others	281	154	54.8%	44	15.7%	83	29.5%
No of previous episodes ^#^ (N, %)	>3	230	130	56.5%	33	14.3%	67	29.1%
	1–3	98	57	58.2%	20	20.4%	21	21.4%
	0	11	8	72.7%	0	0.0%	3	27.3%
Antidepressant ^#^ (N, %)	YES	99	60	60.6%	18	18.2%	21	21.2%
	NO	239	134	56.1%	35	14.6%	70	29.3%
Antipsychotics other than clozapine ^#^ (N, %)	YES	309	171	55.3%	50	16.2%	88	28.5%
	NO	28	23	82.1%	3	10.7%	2	7.1%
Clozapine ^#^ (N, %)	YES	108	70	64.8%	10	9.3%	28	25.9%
	NO	225	122	54.2%	41	18.2%	62	27.6%
Lithium ^#^ (N, %)	YES	21	17	81.0%	2	9.5%	2	9.5%
	NO	317	177	55.8%	51	16.1%	89	28.1%
Benzodiazepines ^#^ (N, %)	YES	190	109	57.4%	24	12.6%	57	30.0%
	NO	148	85	57.4%	29	19.6%	34	23.0%
Anticonvulsant ^#^ (N, %)	YES	79	45	57.0%	13	16.5%	21	26.6%
	NO	258	149	57.8%	39	15.1%	70	27.1%
Failed Antipsychotics ^#^ (N, %)	≥3	220	132	60.0%	29	13.2%	59	26.8%
	1–2	102	55	53.9%	19	18.6%	28	27.5%
	0	14	6	42.9%	4	28.6%	4	28.6%
ECT type ^#^ (N, %)	Bifrontal	274	163	59.5%	37	13.5%	74	27.0%
	Bitemporal	48	25	52.1%	9	18.8%	14	29.2%
	Right unilateral	17	8	47.1%	6	35.3%	3	17.6%

^#^ With missing data.

**Table 2 brainsci-12-00545-t002:** Path model: Direct and indirect effects of ECT induced negative symptoms change on global functioning and cognition function.

		*r*	*se*	*p* Value
From:		to Change of GAF		
Change of positive symptoms	direct effect	−0.611	0.108	<0.001 **
Change of negative symptoms	direct effect	−0.496	0.152	0.001 **
Change of depressive symptoms	direct effect	−0.491	0.152	0.001 **
Change of manic symptoms	direct effect	−0.681	0.196	0.001 **
From:		to Change of MoCA		
Age	direct effect	−0.084	0.040	0.034 *
Female gender (vs. male)	direct effect	2.804	1.117	0.012 *
Change of GAF	direct effect	0.155	0.057	0.007 **
Change of positive symptoms	indirect effect	−0.095	0.038	0.014 *
Change of negative symptoms	indirect effect	−0.077	0.037	0.035 *
Change of depressive symptoms	indirect effect	−0.076	0.036	0.034 *
Change of manic symptoms	indirect effect	−0.106	0.050	0.035 *
Model fit: χ^2^, *p* = 0.278; RMSEA = 0.03; CFI = 0.985; SRMR = 0.045				

* *p* < 0.05, ** *p* < 0.01. Abbreviations: r: correlation coefficient; se, standard error; RMSEA, Root Mean Square Error of Approximation; CFI, Comparative Fit Index; SRMR, Standardized Root Mean Square Residual; GAF, Global Assessment of Functioning; MoCA, Montreal Cognitive Assessment.

**Table 3 brainsci-12-00545-t003:** Path model: Predictors of ECT induced negative symptoms improvement.

		*r*	*se*	*p* Value
From:		To: Own consent		
Voluntary admission status (vs. involuntary)	direct effect	0.276	0.038	<0.001 **
From:		To: Negative symptoms improvement		
Own consent (vs. consent by others)	direct effect	0.135	0.06	0.025 *
With lithium (vs. without lithium)	direct effect	0.252	0.152	0.006 **
Baseline negative symptoms score	direct effect	0.067	0.006	<0.001 **
Voluntary admission status (vs. involuntary)	indirect effect	0.037	0.017	0.032 *
Model fit: χ^2^, *p* = 0.183; RMSEA = 0.043; CFI = 0.990; SRMR = 0.027				

* *p* < 0.05, ** *p* < 0.01. Abbreviations: r: correlation coefficient; se, standard error; RMSEA, Root Mean Square Error of Approximation; CFI, Comparative Fit Index; SRMR, Standardized Root Mean Square Residual.

## Data Availability

The data that support the findings of this study are available from the Electronic Health Records of Neurostimulation service lab, Institute of Mental Health but restrictions apply to the availability of these data, which were used under license for the current study, and not publicly available. Deidentified data are, however, available upon reasonable request and with permission of the, Institutional Research Review Committee and the National Healthcare Group Domain Specific Review Board.

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
