# Peer review of "The Impact of Electroconvulsive Therapy on Negative Symptoms in Schizophrenia and Their Association with Clinical Outcomes"

_brainsci, 2022, doi:10.3390/brainsci12050545_

Round 1
Reviewer 1 Report
This is a well written paper on an original and clinically important subject.
However, I hope the authors would address a few minor issues.
First, I hope the authors can specify what part of the bprs they use to determine change in negative symptoms. Since the bprs has limited capacity to assess negative symptoms.
Second, the authors show that the use of lithium also reduces negative symptoms. Hopefully, the authors can explain why lithium could contribute to the improved outcome. Moreover, hopefully the authors can explain why this effect of lithium is based on a limited number of patients, i.e., only 21 out of the 340 patients used lithium.
Third, the improvement of negative symptoms might seem limited. i.e. compared to the reduction of positive symptoms.
Perhaps the authors could further elaborate on the significance of the found decline of negative symptoms for instance when compared to a reduction of negative symptoms when medication is being used.
Finally, perhaps the authors could explain why they used three different types of ect placements and also used different pulse widths.
Author Response
This is a well written paper on an original and clinically important subject.
However, I hope the authors would address a few minor issues.
First, I hope the authors can specify what part of the bprs they use to determine change in negative symptoms. Since the bprs has limited capacity to assess negative symptoms.
-We thank reviewer’s comments. Specifically, the negative symptoms were measured by a summed score of BPRS item 14. disorientation, item 16. blunted affect, item 17. emotional withdrawal and item 18. motor retardation according to Burlingame G et al 2006. We have added this part and cited the reference paper into main text (revised manuscript, page 7, line 158-160)
Second, the authors show that the use of lithium also reduces negative symptoms. Hopefully, the authors can explain why lithium could contribute to the improved outcome. Moreover, hopefully the authors can explain why this effect of lithium is based on a limited number of patients, i.e., only 21 out of the 340 patients used lithium.
-We discussed in the manuscript that lithium stood out as a factor to predict ECT induced negative symptoms improvement. There is limited literature on the effect of lithium in patients undergoing ECT. The mechanism of lithium to be associated with ECT induced negative symptoms outcome remains unclear, but might be due to the neurophysiological effect of lithium via lowering the electroshock induced seizure thresholds.(Roy and Mukherjee, 1982, Schmidt, 1986) Correlation of lithium with negative symptoms response may also be attributed to the cognitive disturbance caused by a combination of ECT and lithium prescription.(Patel et al., 2020)
Our study is a retrospective analysis of sample patients who had gone ECT treatment. Lithium was prescribed depending on the patient’s illness condition which was happened to be 21/total 340 patients. Thus, our finding regarding role of lithium is limited by the statistic power of 21 patients and future studies with larger sample size or prospective randomized study design may help to validate our observation regarding the moderating role of lithium on negative symptoms improvement.
Third, the improvement of negative symptoms might seem limited. i.e. compared to the reduction of positive symptoms.
Perhaps the authors could further elaborate on the significance of the found decline of negative symptoms for instance when compared to a reduction of negative symptoms when medication is being used.
-We agree with reviewer. In our study, we found the ECT induced negative symptoms improvement on average was small compared to the reduction of positive symptoms. One possible explanation is that most of the study population of schizophrenia for clinical ECT treatment including ours in Singapore (Tor, et al, 2019) are patients with dominant positive symptoms (catatonia is the next). So similar to depressive symptoms in schizophrenia, the negative symptoms improvement induced by ECT may not be specific but a secondary effect of the treatment of positive symptoms. (Zierhut, Marco M et al, 2021) Future ECT studies of patients with dominant negative symptoms may help to clarify this.
However, as we have discussed, so far there is no good medication shows specific effect on negative symptoms of schizophrenia. While the effect of ECT on negative symptoms appeared smaller compared to other symptom domains, we did observe subgroup of patients with significant negative symptoms improvement. Identifying and characterise this subgroup of patients who might benefit from ECT treatment could shed light on the precision utilization of ECT in the treatment of negative symptoms. We have rephrased this section to be clearer (line 382-384).
Finally, perhaps the authors could explain why they used three different types of ect placements and also used different pulse widths.
Treatment guidelines vary in their recommendations for the use of ECT in schizophrenia. In our centre, we adopt different types of ECT placement and pulse widths according to the reports of several randomized clinical trials for schizophrenia patients and mechanism studies of ECT (Petrides, G., et al. 2015, Phutane, V. H., et al. 2013; Krystal, A. D., et al. 1998). The treatment efficacy of the adopted guidelines in Singapore population had been retrospectively analysed and reviewed. (Chan CYW et al. 2019.) Other than the guidelines, in real-life practise, the ECT parameters were sometimes decided by treating psychiatrist after a complex evaluation of the clinical scenario and patient characteristics which include patient age, patients’ preference, past response of ECT and cognitive side effect, duration of illness and severity of current episode etc.

Reviewer 2 Report
In this manuscript, the authors investigated the effect of ECT on negative symptoms in schizophrenia and factors that moderate this effect. The sample size is big and the research question is interesting and has potential clinical implications. The data analysis, however, is poorly conducted, in its current version, the path analysis part is confusing.
Major concerns:
1, introduction: can the authors discuss why previous studies with ECT have generated inconsistent results, and what is the novelty of the current study?
2, data reporting in Table 2: given this is the first main result, can the authors provide a visual illustration of the individual changes from pre to post ECT for all measurements? for instance, using figures like this (https://twitter.com/jordyvanlangen/status/1271096169926660099) or fig 3 of this paper (https://www.sciencedirect.com/science/article/pii/S0095447017301407)
3, Figures 1 and 2, and Tables 3 and 4 are confusing and should be removed according to the reviewer's point of view. The authors aimed to investigate what factors affect the ECT-induced changes in negative symptoms. For such purpose, what they need to do is 1) conduct a correlation analysis between the variables (or t test for gender), 2) conduct a multiple regression analysis to predict the ECT-induced changes in negative symptoms. To better understand the associations, they should also provide a scatterplot to let readers see the data directly.
4, sample size issue: section 3.1 total sample size is 340; table 2 sample size for MoCA is 211; however, the sample size in section 3.3 for studying the correlation between changes in symptoms and cognition is 275. Can the authors confirm this inconsistency?
Author Response
In this manuscript, the authors investigated the effect of ECT on negative symptoms in schizophrenia and factors that moderate this effect. The sample size is big and the research question is interesting and has potential clinical implications. The data analysis, however, is poorly conducted, in its current version, the path analysis part is confusing.
Major concerns:
1, introduction: can the authors discuss why previous studies with ECT have generated inconsistent results, and what is the novelty of the current study?
-Data inconsistence is very common in both research trials and clinical treatment. Even with controlled randomized clinical trial study design, the investigators may sometimes report inconsistent results due to lots of unknown confounders including the heterogeneity of study population, variety of comorbid intervention, different observation period and statistic analysing methods etc.
As we have introduced, in this study, we aimed to examine the effect of ECT on not only negative symptoms but also the correlation of ECT induced negative symptoms change with change of cognition and general functioning. (line 122-124) Further, we aim to identify predictors of negative symptoms improvement due to ECT treatment. Identifying and characterise this subgroup of patients who might benefit from ECT treatment could shed light on the precision utilization of ECT in the treatment of negative symptoms. (Line 384-386).
2, data reporting in Table 2: given this is the first main result, can the authors provide a visual illustration of the individual changes from pre to post ECT for all measurements? for instance, using figures like this (https://twitter.com/jordyvanlangen/status/1271096169926660099) or fig 3 of this paper (https://www.sciencedirect.com/science/article/pii/S0095447017301407)
-Changed to figure 1 as requested
3, Figures 1 and 2, and Tables 3 and 4 are confusing and should be removed according to the reviewer's point of view. The authors aimed to investigate what factors affect the ECT-induced changes in negative symptoms. For such purpose, what they need to do is 1) conduct a correlation analysis between the variables (or t test for gender), 2) conduct a multiple regression analysis to predict the ECT-induced changes in negative symptoms. To better understand the associations, they should also provide a scatterplot to let readers see the data directly.
-We do not agree with the reviewer. Path analysis using structured equation modelling is a powerful and commonly used method of statistical analysing basing on correlation matrix between any two variables in the model. Path analysis is an extended multiple regression but path analysis can be used to analyse models that are more complex (and realistic) than multiple regression as we have no predefined factors that can be used to predict negative symptoms change. Moreover, it can simultaneously analysis multiple moderators/mediators and compare different models to determine which one best fits the data, which is unlikely to be coped with by multiple regression analysis. Path analysis can also disprove a model that postulates causal relations among variables although it cannot prove causality. (Streiner 2005) More specifically, in our study, negative symptoms change is an outcome of ECT treatment which occurs after the patient characteristics and treatment parameters have been fixed while regression analysis is a simple correlation analysis between factors which reflecting no causal relationships. Thus, the interpretations of path analysis results and regression analysis results are different. We have added this part into manuscript and new references with similar study design were cited (line 185-192).
4, sample size issue: section 3.1 total sample size is 340; table 2 sample size for MoCA is 211; however, the sample size in section 3.3 for studying the correlation between changes in symptoms and cognition is 275. Can the authors confirm this inconsistency?
-In our study, the total number of patients is 340. However, as we have discussed in the manuscript (line 367-369), patients who had been referred to ECT treatment are generally very ill. Therefore, some patients are with missing MoCA assessment or their condition was too ill for MoCA assessment either at before or after ECT treatment. When we do t test to compare the pre vs post ECT treatment score, (original table 2, revised figure 1) we only count patients with complete set of MoCA assessment (n=211). In Section 3.3, total 270 (not 275, apologize for the mistake) patients were included for correlation analysis between changes in symptoms and cognition as SEM analysis can include patients (observations) with missing values to certain extent, depending on the pattern and strength of correlation matrix.

Round 2
Reviewer 2 Report
Thank the authors for addressing my concerns. However, I disagree with the authors that path analysis is useful for the primary purpose of the study, more direct parsimonious methods such as correlation with scatterplots and regression analysis are preferred according to the reviewer's point of view. Sorry.